# Single-View Measurement Method for Egg Size Based on Small-Batch Images

**DOI:** 10.3390/foods12050936

**Published:** 2023-02-22

**Authors:** Chengkang Liu, Qiaohua Wang, Meihu Ma, Zhihui Zhu, Weiguo Lin, Shiwei Liu, Wei Fan

**Affiliations:** 1College of Engineering, Huazhong Agricultural University, Wuhan 430070, China; 2Ministry of Agriculture Key Laboratory of Agricultural Equipment in the Middle and Lower Reaches of the Yangtze River, Wuhan 430070, China; 3National Research and Development Center for Egg Processing, Huazhong Agricultural University, Wuhan 430070, China; 4College of Food Science and Technology, Huazhong Agricultural University, Wuhan 430070, China

**Keywords:** actual outline of eggs, deep learning, egg-carrying component, image segmentation, single-view metrology

## Abstract

Egg size is a crucial indicator for consumer evaluation and quality grading. The main goal of this study is to measure eggs’ major and minor axes based on deep learning and single-view metrology. In this paper, we designed an egg-carrying component to obtain the actual outline of eggs. The Segformer algorithm was used to segment egg images in small batches. This study proposes a single-view measurement method suitable for eggs. Experimental results verified that the Segformer could obtain high segmentation accuracy for egg images in small batches. The mean intersection over union of the segmentation model was 96.15%, and the mean pixel accuracy was 97.17%. The R-squared was 0.969 (for the long axis) and 0.926 (for the short axis), obtained through the egg single-view measurement method proposed in this paper.

## 1. Introduction

Eggs are an inexpensive and nutritious food, and egg quality grading is very important in commercialization. Furthermore, the measurement accuracy of egg size is essential in quality testing and affects the hatching rate of breeder eggs [1,2]. However, manual labor cannot consistently perform the detection perfectly because of eye strain. A machine vision approach would not suffer from fatigue, but failure to capture an actual outline view results in measurement errors due to the ellipsoidal appearance of eggs, which affects the determination of the eggs’ appearance quality. Therefore, it is crucial to study how to obtain the exact size of eggs through machine vision.

Many scholars used machine vision for egg size detection, volume prediction, and freshness grading. Among them, some scholars used external parameters such as egg size to achieve the primary grading. Omid developed an intelligent system based on machine vision to achieve egg grading using defects and size [3]. Vasileva analyzed the external parameters of eggs through machine vision to automate the primary grading stage [4]. Some scholars achieved egg detection through quantifying volume and appearance parameters, where volume is estimated using images. Soltani proposed a new method for egg volume prediction and implemented an efficient image processing algorithm for egg size measurement [5]. Quilloy developed an automatic single-line tabletop egg sorter that integrated machine learning and electromechanical principles; the sorter used egg projection area and weight estimation, which is more efficient than manual sorting [6]. Harnsoongnoen performed egg grading and freshness assessment based on machine vision and load cells; they used the machine vision system to estimate the eggs’ volume, and the results indicate that the system has potential in the poultry industry [7]. In addition, machine vision has been applied to breeder eggs. Haixia detected egg size through machine vision and found that the detection values highly correlated with the measured values [1]. Xiulian achieved the rejection of sperm-less eggs through machine-vision-extracted regional invariance parameters such as the long-to-short axis ratio [8]. Weizhong screened breeder eggs based on machine vision, using egg size as indicators, and the screening results were highly consistent with manual results [2]. Okinda used an infrared depth sensor to estimate the volume of eggs on the production line and developed a volume estimation model using feature variables [9]. Nasiria achieved the classification of unwashed eggs using machine vision and deep learning methods [10]. Aboonajmi discussed the importance of using non-destructive inspection methods such as machine vision in an in-line sorting system for the internal and external quality of eggs [11]. Valeriy proposed a mathematical formula for describing the shape of the egg based on four parameters, namely egg length, maximum breadth, the shift of the vertical axis, and the diameter at one-quarter of the egg length [12]. Liao proposed a clear definition of the standard egg shape curve in an analytical geometric sense and also discovered the equation for the segmental function of the curve [13]. All the literature above used machine vision to achieve egg quality detection, but the two following issues are overlooked in acquiring egg images: first, the imaging plane’s height differs for different egg sizes; second, due to eggs’ ellipsoidal appearance and the component’s structure, the imaging outline may not match the actual outline of the egg.

Single-view metrology is convenient and effective for studying object dimensioning through images. Some scholars studied theoretical and algorithmic geometric information extraction for single uncalibrated images [14]. These works focused on new single-view metrology, including new methods for obtaining vanishing points [15,16], off-surface point transfer [17], single-view metrology after lens distortion correction [18], and calculating camera matrix parameters using vanishing points [19]. Other scholars studied applications for single-view metrology, including road horizontal and vertical measurements [20], distance estimation in the car driving [21], Google Street View building height estimation [22], and real-time target tracking and manipulation combined with robotics [23]. Single-view metrology requires specific geometric information based on known scenes. The measurement accuracy depends heavily on the accuracy of image preprocessing, such as edge detection, line fitting, and vanishing point determination [19]. This paper proposes a measurement method for eggs based on the above single-view metrology, which solves the problem of current machine vision methods being unable to capture the actual outline of eggs. In addition, this paper obtained an egg image segmentation model by combining small-batch egg samples with deep learning. Small-batch eggs refers to using a small number of egg samples to obtain the egg segmentation model, which can obtain the segmented image of a single egg; this model combined with single-view metrology can achieve egg size measurement, laying the foundation in egg commercialization lines.

The goal of this study is to achieve size measurement of small-batch eggs through single-view metrology and deep learning. The specific studies are (1) designing the egg-carrying component to realize the egg’s outline acquisition in a single view, (2) achieving image segmentation for eggs in small batches through deep learning, and (3) designing the geometric environment for vanishing point acquisition based on single-view metrology. Moreover, we completed the mathematical measurement method applicable to the egg’s major and minor axes.

## 2. Materials and Methods

### 2.1. Experimental Materials

A total of 30 eggs under the Sundaily Farm brand were purchased (Wuhan, China), and the mean and median weights of all samples measured using an electronic scale were 47.65 g and 46.71 g, respectively. The eggs’ major axis and minor axis were measured using vernier calipers. In addition, egg sizes were measured five times in the morning and afternoon using vernier calipers to reduce random errors. After the vernier caliper measurement, a camera (IDS company, Saarbrücken, Germany, CCD industrial camera, model UI-2210RE-C-HQ, resolution 640 × 480; lens model M0814-MP2) and a surface lamp (Guiguang Instrument company, Guilin, China, DY30-1C, AC220V, DC24V, 30 W) were used to collect the egg images (Figure 1).

As in Figure 2a,b, the egg deflected to different degrees in different views when placed on a flat surface. This placement causes the problem that the egg’s major and minor axes cannot be measured using a single view. The egg-carrying component (Figure 2c,d) designed in this paper ensured that the major axis was parallel to the front view of the egg. Thus, the egg-carrying component designed in this paper can help in obtaining the actual outline of the egg from a single view. When the egg slides down from the top of the component, it adjusts its posture using gravity. The egg-bearing component was obtained via 3D printing using resin material; the component was 75 mm long, 64 mm wide, and 34 mm high. The principle is to use the tangency relationship between the egg’s cross-sectional circles and the component’s triangles boundary. Here, Table 1 shows the differences between this component and existing egg image capture components.

### 2.2. Single-View Measurement for Eggs

(1)Single-View measurement principle

The length, width, and height properties of three-dimensional objects cannot be measured from a single view alone, but the length ratio of two parallel objects can be obtained. Thus, if a reference object is added and the actual size is known, combined with the length ratio, the size of the measured object can be obtained. Criminisi described how the affine 3D geometry of a scene could be measured from a single perspective image [14]. The proposed basic geometry of a single view is shown in Figure 3. The vanishing line *l* is the projection of the line at infinity of the reference plane in the image. The vanishing point *v* is the image of the point at infinity in the reference direction.

Criminisi proposed interplanar distance measurement [24]; this method (Figure 4) uses the cross-ratio with a known distance $Z_{r}$ (the distance between the planes *π* and $\pi^{\prime }$) to obtain the value of *Z* (the distance between the planes $\pi_{r}$ and $\pi_{r}^{\prime }$). First, the mathematical relation between the distance $Z_{c}$ and *Z* can be established according to the cross-ratio. $Z_{c}$ is the distance of the camera from the plane *π*.
(1)$$\frac{D\left(x,c\right)}{d\left(x^{\prime },c\right)}/\frac{d\left(x,v\right)}{d\left(x^{\prime },v\right)}=\frac{d\left(X,C\right)}{d\left(X^{\prime },C\right)}/\frac{d\left(X,V\right)}{d\left(X^{\prime },V\right)}$$

X is a point on the plane π, $X^{\prime }$ is the parallel projection of the point X on the plane $\pi^{\prime }$, and d() is the distance between the two points. Since the vanishing point *V* is at infinity, $\frac{d\left(X,V\right)}{d\left(X^{\prime },V\right)}=1$, it follows that,
(2)$$\frac{Z}{{Z\;-\;Z}_{c}}=\frac{d\left(X,C\right)}{d\left(X^{\prime },C\right)}=\frac{d\left(x,c\right)d\left(x^{\prime },v\right)}{d\left(x^{\prime },c\right)d\left(x,v\right)}$$

In order to compute the distance *Z*, it is necessary to determine $Z_{c}$ by the distance $Z_{r}$. According to the cross-ratio, the mathematical relation between $Z_{cr}$ (the distance of the camera from the plane $\pi_{r}$) and $Z_{r}$ (the distance between the planes $\pi_{r}$ and $\pi_{r}^{\prime }$) can be obtained as follows,
(3)$$\frac{Z_{r}}{Z_{\mathrm{cr}}}=1\;-\;\frac{d\left(r_{1},v\right)d\left(r_{2}{,c}_{r}\right)}{d\left(r_{2},v\right)d\left(r_{1}{,c}_{r}\right)}$$
$r_{1}$ and $r_{2}$ are the points corresponding to $R_{1}$ and $R_{2}$ in the image, and $c_{r}$ is the intersection of the lines $r_{1}$ and $r_{2}$ with the fading line *l*. Where $Z_{r}$ is known, the distance $Z_{cr}$ can be determined. Further, based on the cross-ratio and the distance ($Z_{c}{\;-\;Z}_{cr}$) between the plane $\pi_{r}$ and π, the distance $Z_{c}$ can be determined as follows,
(4)$$Z_{c}{=Z}_{\mathrm{cr}}\frac{d\left(s_{1}{,c}_{s}\right)d\left(s_{1},v\right)}{d\left(s_{2}{,c}_{s}\right)d\left(s_{2},v\right)}$$

Further, the distance *Z* can be computed by combining Equation (4) with Equation (2).

(2)Vanishing point determination and measurement principle of egg images

In this paper, the vanishing point was determined using the multi-vanishing point determination method proposed by Nietoa [25]; the method is based on a robust version of the MLESAC (maximum likelihood estimation by sample and consensus) algorithm. Moreover, it is used similarly to the EM (expectation maximization algorithm), which allows a trade-off between accuracy and efficiency and enables real-time manipulation of video sequences. Nietoa provides the source code for download at https://sourceforge.net/projects/vanishingpoint/ (accessed on 5 July 2022) for the figures that were drawn in this paper. As shown in Figure 5, we prepared the cards with crossed parallel lines to obtain vanishing points.

This paper used vanishing lines and points to measure the egg’s major and minor axis based on the interplanar distance measurement principle proposed by Criminisi. First, as shown in Figure 6a, geometrizing the egg and the reference as a straight line, this paper assumed that *AB* is the egg of length *Z* and *CD* is the reference of length *R*, where the projection of *CD* on *AB* is *FB*. Next, as shown in Figure 6b, the parallel lines of the reference plane were used to obtain the vanishing points $v_{l}$ and $v_{r}$. The vanishing lines were obtained by connecting the two vanishing points, and all the lines from the vanishing lines are parallel. Therefore, *bd* intersects the vanishing line at *e*, and *ec* then intersects *ab* to obtain the projection point *f*. Based on the cross-ratios, the mathematical relation between the geometrized egg and the reference can be obtained as follows,
(5)$$\frac{d(B,\;F)}{d(B,A)}/\frac{d(G,F)}{d(G,A)}=\frac{d(b,f)}{d(b,a)}/\frac{d(g,f)}{d(g,a)}$$

Since point *G* is at infinity, $\frac{d(G,F)}{d(G,A)}=1$, Equation (5) can obtain the following equation.
(6)$$\frac{R}{Z}=\frac{d\left(b,f\right)d\left(g,a\right)}{d\left(b,a\right)d\left(g,f\right)}$$
R is the reference height, a known quantity, so the length Z can be computed by combining the image processing results with Equation (6).

### 2.3. Egg Image Segmentation in Small Batches

In order to achieve accurate and efficient image segmentation, this paper used Segformer to complete the egg image segmentation. Then, the egg size was measured based on single-view metrology. Segformer is a simple but powerful semantic segmentation method proposed by Xie; the method avoids the complex design commonly found in previous methods. Furthermore, it unifies the transformer encoder with a new hierarchy that does not require positional encoding and the lightweight multilayer perceptron (MLP) decoder. Thus, it achieves high efficiency and performance and shows zero-shot robustness [26]. In this paper, the weights provided by Xie (pre-trained on ImageNet-1K) were used. Fifteen samples were randomly selected from thirty data samples as the test set to ensure independence. The remaining 15 samples were expanded to 75 for training by randomly flipping up and down, flipping left and right, and varying contrast and lightness. Moreover, the specific training parameters are shown in Table 2. Annotation was performed using the PixelAnnotationTool (https://github.com/abreheret/PixelAnnotationTool, accessed on 15 June 2022) to obtain labels, and then trained and tested based on the code and usage provided by Enze (https://github.com/NVlabs/SegFormer, accessed on 28 July 2022).

### 2.4. Single-View Measurement Framework and Evaluation Metrics

The proposed measurement framework consists of three main steps to measure the egg’s major and minor axis (Figure 7). The first step is to acquire a single-view image using the component that can be calibrated for the egg pose. The second step is to obtain segmentation images of eggs and references through a Segformer model trained with a few-shot. The third step is to obtain the vanishing point or the egg’s major and minor axis through egg-size single-view metrology.

In this paper, we used intersection over union (IoU) and pixel accuracy (PA) as the performance discriminators of the segmentation model.
IoU = TP/(TP + FP + FN)(7)
PA = (TP + TN)/(TP + TN + FP + FN)(8)

Here, TP indicates that the model predicts a positive case and predicts correctly, FP indicates that the model predicts a positive case and predicts incorrectly, FN indicates that the model predicts a negative case and predicts incorrectly, and TN indicates that the model predicts a negative case and predicts correctly.

The absolute error was used as an evaluation index to confirm the performance of the single-view measurement framework. Furthermore, R^2^ (R-squared) was used to assess the conformity degree between the single-view measured value and the actual value. In particular, to eliminate random errors in the measurement of actual values, each egg was measured by the same experimenter five times in the morning as well as the afternoon. Then, the average of the measured values was taken.

## 3. Results

### 3.1. Segmentation Result of Egg Images

The mean intersection over union (MIoU) of the segmentation model was 95.35%, and MPA (pixel average accuracy) was 97.31%. IoU and PA for each category are shown in Table 3. Table 3 shows that the model has the best segmentation result for the background, with both IoU and PA reaching over 99%. The segmentation result of the egg is fine, with IoU and PA reaching 97% or more. The reference has the worst segmentation result, with a PA of 89.22% and Iou of 93.71%. It can be seen that the Segformer algorithm demonstrated strong segmentation performance. However, the small area in the image, the similar color to the background, and being occluded (Figure 8) resulted in poor segmentation of the reference. Therefore, to reduce the measurement error caused by reference segmentation, the average of the reference coordinate values (the egg size calculation requires the reference coordinates, as detailed in Section 2.2, Figure 6) from multiple images was taken for the egg size calculation.

### 3.2. Vanishing Point Image

Figure 9 shows the vanishing points obtained through the multiple vanishing point determination method (Section 2.2). Since the egg’s ideal major and minor axes are perpendicular to different planes, the original image was rotated by 90 degrees to obtain the vanishing points of the right card. In addition, we cropped the images to prevent the two cards from interfering with each other in acquiring the vanishing points; we supplemented the vanishing point coordinate values in the measurement calculations.

### 3.3. Single-View Measurement Result of Egg Size

Table 4 shows that the measurement errors of the major axis were all less than 1 mm, and the adj.R^2^ was 0.9725 (Figure 10), which was obtained from the regression analysis of the measured and actual values. For the minor axis, the measurement errors of four samples were greater than 1 mm, but they were all within 2 mm, and the adj.R^2^ was 0.8353 (Figure 10). In addition, this paper used an F-test for the regression models, and the results show that all models have a significant linear relationship (Table 5).

The measurement results of the minor axis are undesirable. The possible reasons are as follows; first, the segmentation of the reference object could be more satisfactory, where the length and width determine the measurements of the major and minor axes of the egg. In addition, Figure 8 shows that there is more interference from the same color background towards the short side of the reference object, so it leads to a greater calculation error of the minor axis. Second, observing Figure 8, the contact position of the egg with the egg-bearing component is very dark, which also interferes with the segmentation of the short axis direction of the egg, which affects the calculation of the minor axis.

## 4. Discussion

In this paper, we designed the egg-bearing component to acquire the egg’s actual outline. This component uses the tangency of the egg’s cross-sectional circle to the triangular boundary; we found that this component provides an excellent actual outline of eggs by comparing the actual and measured values of the egg’s major and minor axis. However, due to the ellipsoidal shape of the egg, the component can only obtain the actual outline of the front view and not the actual outline of the other views.

Most of the current studies into egg detection via image have been performed by processing and measuring directly through the image. Additionally, this research did not consider the consistency of the egg imaging plane. However, it is vital to ensure that the image plane of the egg is consistent. The literature [27] also suggests that whether the major axis is parallel to the horizontal plane affects the measurement of eggs, and the literature agrees that a suitable egg roller can improve the accuracy of egg measurements. Compared with these studies [7,28,29], the method presented in this study can disregard the consistency of the egg imaging plane by applying single-view metrology. Using the vanishing point in the perspective view and the reference size, the difference in the imaging plane does not affect the measurement for different sizes of eggs.

However, the measurement accuracy of the egg sizes depends on image processing. In this paper, an image segmentation model was developed using Segformer. Segformer can build a robust image segmentation model with few samples, but the image background is the same color as the reference; we obtain a worse reference segmentation. Therefore, we will use different colored backgrounds or references in future studies. This paper verified the feasibility of single-view metering eggs using an egg-carrying component and single-view metrology. Future research should explore a dynamic egg-carrying component. The dynamic online measurement component can provide a reliable and efficient basis for determining the eggs’ appearance quality in the commercialization process. In addition, the component can provide researchers with a dynamic acquisition method for single-view dimensional metrology.

Taken together, our results show that the actual outline of eggs can be obtained using the tangency of the egg cross-section circle to the triangular boundary; single-view metrology can solve the problem of inaccurate measurements caused by different egg imaging planes. However, this paper only verified the method’s feasibility using the static egg-bearing component. Moreover, further improvements in the image segmentation method are yet to be made to improve the measurement accuracy of egg size.

## 5. Conclusions

This paper verified the feasibility of single-view measurement of the egg’s major and minor axes; this paper proposed the method of single-view egg major and minor axis determination by combining with single-view metrology. Due to the strong zero-shot robustness and high efficiency of Segformer, this paper achieved good image segmentation using a small batch of images. Moreover, we obtained excellent measurement results of the eggs’ major and minor axes using the projection relation between parallel lines and the vanishing line property. In order to simplify model building and validation, this work chose a single type of egg for experiments, and subsequent exploration of universal models for multiple types will be undertaken. Future research will focus on multiple types of egg measurements and online dynamic egg measurements.

## Figures and Tables

**Figure 1 foods-12-00936-f001:**
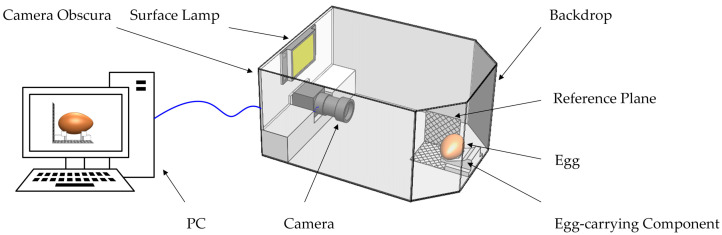
Schematic diagram of egg image acquisition.

**Figure 2 foods-12-00936-f002:**
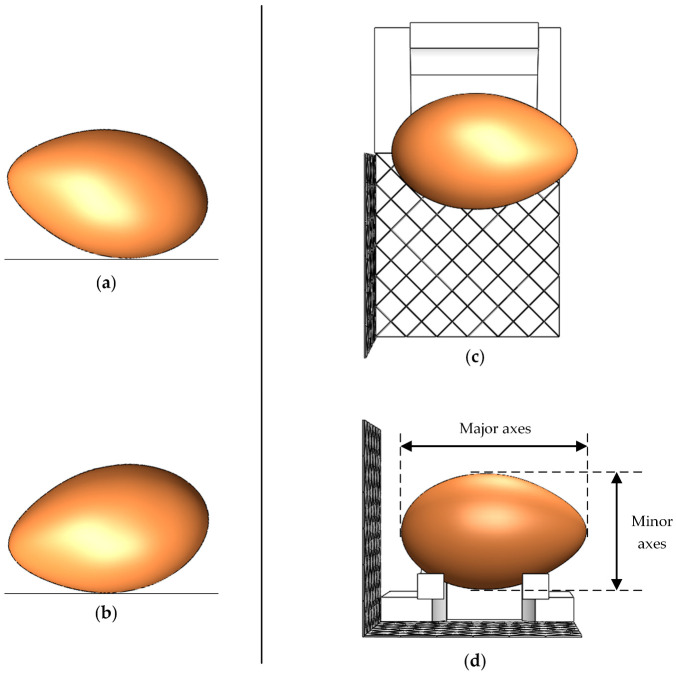
Different view of the egg (possible poses placed naturally on a flat surface, (**a**) front view; (**b**) top view. The stance using component, (**c**) top view; (**d**) front view).

**Figure 3 foods-12-00936-f003:**
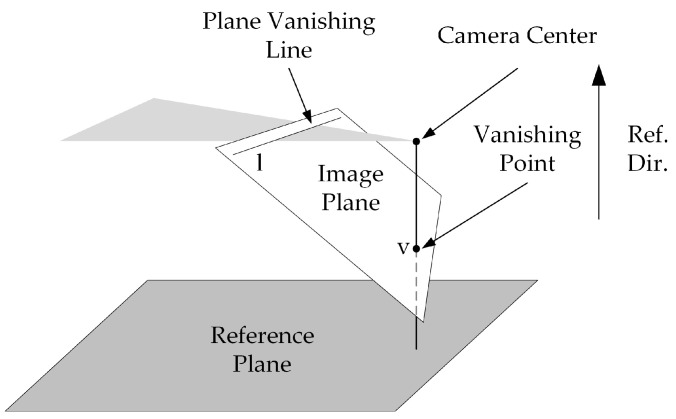
Basic geometry of single-view metrology.

**Figure 4 foods-12-00936-f004:**
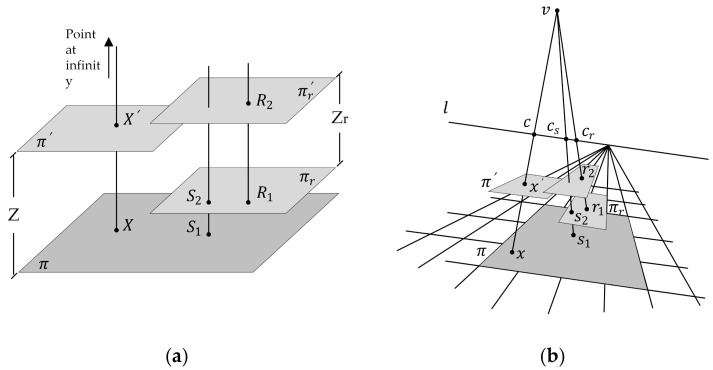
The geometry of calculating plane distance using cross-ratio, (**a**) in the world, (**b**) in the image.

**Figure 5 foods-12-00936-f005:**
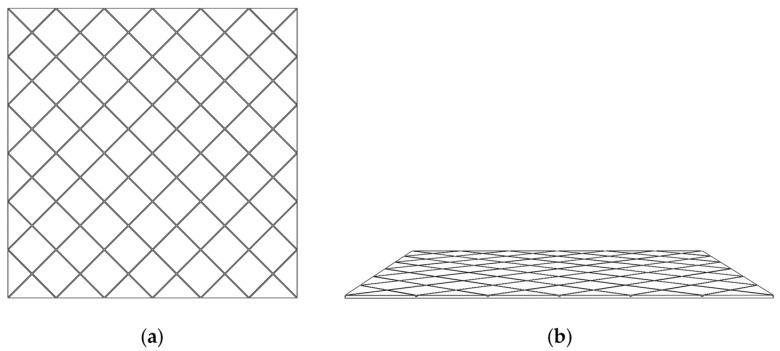
Cross-parallel reference card, (**a**) front view, (**b**) perspective view.

**Figure 6 foods-12-00936-f006:**
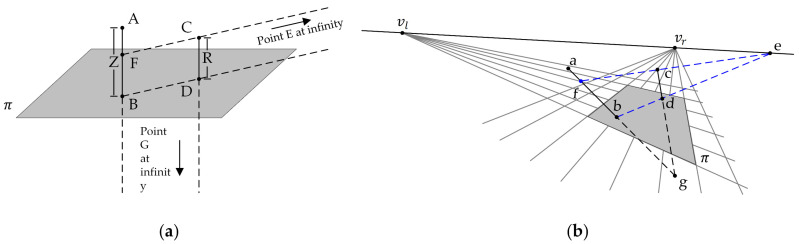
The geometry of obtaining egg size based on cross-ratio and single-view metrology, (**a**) in the world, (**b**) in the image.

**Figure 7 foods-12-00936-f007:**
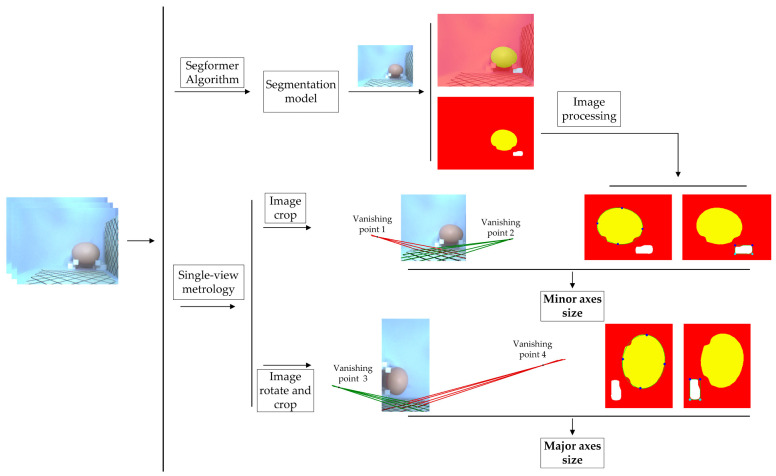
The flow chart of egg single-view metrology.

**Figure 8 foods-12-00936-f008:**
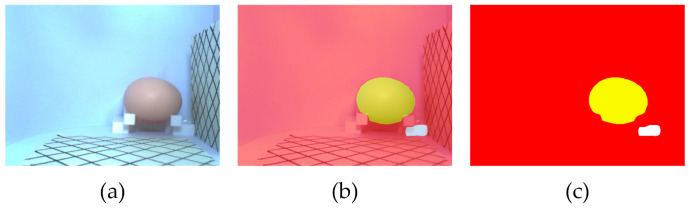
Results of image segmentation. (**a**) Original image; (**b**) image segmentation result; (**c**) mask image.

**Figure 9 foods-12-00936-f009:**
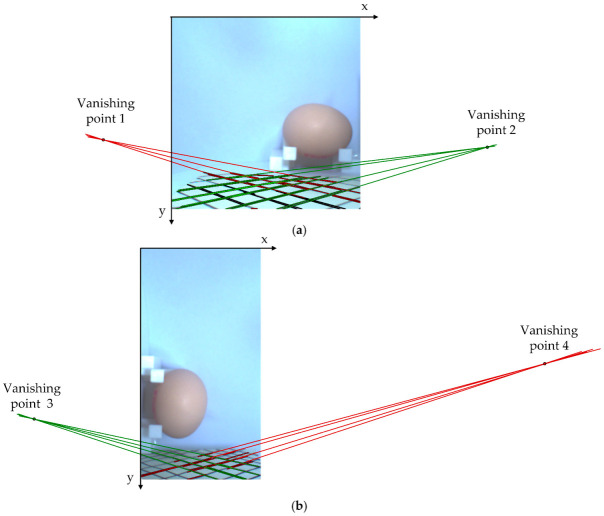
The location of the vanishing points. (**a**) Original image; (**b**) rotated image.

**Figure 10 foods-12-00936-f010:**
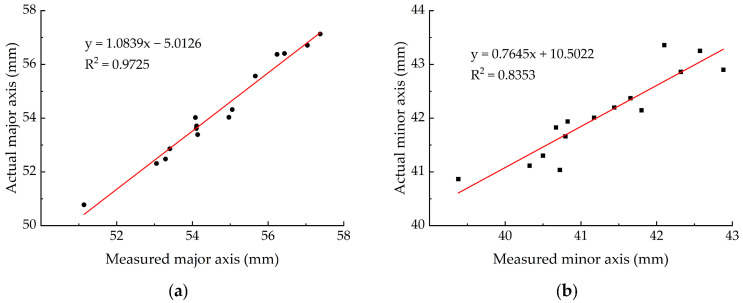
Comparison of measured and actual values of eggs’ size. (**a**) Eggs’ major axis; (**b**) eggs’ minor axis.

**Table 1 foods-12-00936-t001:** Comparison of the egg-carrying component designed in this paper with common egg-carrying methods.

	Egg-Carrying Method	Design Innovation	Advantages
Mahmoud Soltani [5]	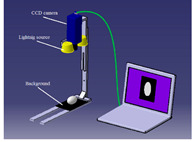	The tangency between the cross-sectional circle of the fresh egg and the triangle of the component is used to obtain the true profile of the fresh egg. The design idea of this component can be used for the adjustment of the fresh egg posture of the production line.	The components designed in this paper allow for a single view to obtain the real outline of the egg and for the adjustment of the egg’s attitude using gravity.
Erwin [6]	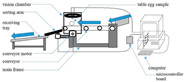		
Harnsoongnoen [7]	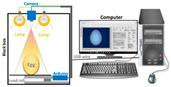		
This paper	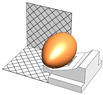		

**Table 2 foods-12-00936-t002:** Parameters used in training model.

Parameter or Resource	Value or Model
Checkpoint_file	pspnet_r50-d8_512×1024_40k_cityscapes_20200605_003338-2966598c.pth
Optimizer	SGD
Batch size	8
Initial learning rate	0.01
Train, validate, and test images	60, 15 and 15 images
Total iterations	5000
Backend	Pytorch 1.7
Python	3.6
Operation system	Windows 10
GPU	NVIDIA RTX 2080 Ti
Class	(‘background’, ‘egg’, ‘ref’)
Palette	(255, 0, 0), (250, 250, 0), (255,255, 255)

**Table 3 foods-12-00936-t003:** Iou accuracy and PA of the segmentation model.

Class	IoU (%)	PA (%)
Background	99.72	99.88
Egg	97.11	98.34
Reference	89.22	93.71

**Table 4 foods-12-00936-t004:** Eggs’ size measurement results.

Serial Number	Minor Axis Size (mm)			Major Axis Size (mm)		
	Measured Value	Actual Value	Error	Measured Value	Actual Value	Error
1	39.329	40.866	1.537	51.136	50.774	0.362
2	40.668	41.036	0.368	57.040	56.713	0.327
3	40.770	41.939	1.169	53.405	52.856	0.549
4	40.742	41.661	0.919	54.140	53.388	0.752
5	40.617	41.828	1.211	53.291	52.478	0.813
6	42.825	42.902	0.077	54.966	54.033	0.933
7	40.445	41.302	0.857	55.053	54.321	0.732
8	41.599	42.373	0.774	54.111	53.712	0.400
9	41.746	42.147	0.401	57.377	57.132	0.245
10	42.517	43.253	0.736	56.433	56.407	0.026
11	42.046	43.360	1.314	56.237	56.376	0.139
12	40.270	41.115	0.845	55.663	55.566	0.097
13	41.385	42.198	0.813	53.052	52.312	0.740
14	41.119	42.011	0.892	54.079	54.024	0.055
15	42.264	42.862	0.598	54.105	53.613	0.492

**Table 5 foods-12-00936-t005:** F-test results for the regression models.

Regression Models	Significance Level	F-Value	
Major axis	0.01	552.999	P_{F ≥ 98.503}_ = 0.01
Minor axis		72.320	P_{F ≥ 34.116}_ = 0.01

## Data Availability

Not applicable.

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
