# Peer review of "Single-View Measurement Method for Egg Size Based on Small-Batch Images"

_foods, 2023, doi:10.3390/foods12050936_

Round 1

Reviewer 1 Report

More recent references should be added to the Introduction.

The novelty of the research should be more clearly indicated on the background of available literature.

The experimental material comprised of a total of 30 eggs belonging to one type is not enough.

The results are not important and useful enough for practical use to be published.

There are no references in the Discussion.

Reviewer 2 Report

This paper proposes an automated method to capture standardized images of eggs with a known reference object, segment the egg with a deep learning approach, and quantify its major and minor axes using single-view metrology. I believe the study is well-done and worthy of publication, but only after a minor revision to improve the quality of the writing and to clarify some experimental details.

Below are some general comments and questions, followed by section-specific edits:

General Comments

Is the coauthor named Shiwei Liu or Shiwei Lin? It is spelled differently in the text vs. in the submission portal.

Throughout the paper, do not refer to other works by the full first and last name of the lead author. For example, line 37 says “Mahmoud Omid developed an intelligent system based on…”. Instead, say something like “Omid et al. [3] developed an intelligent system based on…” This is more consistent with in-text citation style for most journals, including Foods. Please fix this throughout the text.

What does “small-batch” mean in this context? Is this a domain-specific term that has a specific meaning? Unless you think this would be obvious to most readers, please define what you mean by this. An appropriate place to do this might be around line 71, when the term is first introduced, as far as I can tell.

            Related to this point, does only one egg appear in each image? If “small-batch” simply means “a small number of eggs”, then I might find this term misleading since the approach appears to operate on single-egg images. At the very least, please clarify in the text whether this approach is designed for single-egg images only.

More details about Segformer training are needed. What batch size was used? The optimizer? Learning rate? More details about dataset split are also needed. How many images were used for training vs. testing?

How exactly was the segmentation used to perform single-view metrology? I imagine the segmentation was necessary to compute the reference and egg lengths needed to produce the major and minor axis length predictions. Please make this link more concrete, perhaps with an equation or detailed explanation in Section 2.3 or 2.4.

Title

“Small-batch” should be hyphenated to be consistent with its usage elsewhere

Abstract

Line 17: Repetitive to say “This study proposed a single-view measurement method suitable for eggs based on single-view metrology”

Line 18: Should say “Experimental” instead of “Experiment”

Line 20: Should say “intersection over union”

Introduction

Combine first two sentences into one – too terse

Line 30: I don’t quite understand this sentence. First, “machine vision can disregard fatigue” is awkward; I might say “While a machine vision approach would not suffer from fatigue…” Second, I don’t understand how “failure to capture an actual outline view” relates to your point about the importance of an automated machine vision approach.

Line 35-36: I would combine these two sentences or use a transition between the first and second sentence of this paragraph.

Line 40: I don’t know what it means to “achieve egg detection through volume and appearance parameters”. Perhaps “achieved egg detection through quantifying volume and appearance”?

Line 46: Can just say “egg grading”

Line 47: Should be “they” rather than “They”. Don’t capitalize the first word after a semicolon

Line 49: I would say “has been applied” instead of “was applied”

Line 55: I would rewrite to “…the two following issues are overlooked in acquiring egg images: first, the imaging plane’s height…”

Line 61: “These researches” is strange -- I would say “these studies” or “these works”

Line 64: Would say “camera matrix parameters” or “camera matrices” (should be plural how you’ve written it)

Line 67: Unnecessary “The” at beginning of sentence

Line 72: Rewrite to “The specific contributions are (1) designing the egg-carrying component…, (2) achieving image segmentation…, and (3) designing the geometric…”

Materials and Methods

Line 99: I don’t quite agree with this sentence as it’s written currently: “An object cannot be measured from a single view alone.” I would be more specific; presumably you mean that certain properties of 3-dimensional objects cannot be measured from a single view alone.

Line 110: Z_r is the distance from what to what? Also please explain in words what Z represents

Eq 1: What is X’ ?

Line 113: Remove “Where”

Eq 3: What are r_1, r_2, and c_r? What does the function d() represent? I can make reasonable guesses but please make these definitions clear when you introduce new notation. Also ensure that notation is consistent between the text and figures. 

Line 127: “Scheme” does not seem like the appropriate word. Perhaps “version” or “variation”?

Table 1: I would say “Total iterations”

Line 159: Should probably say “zero-shot robustness” or “robustness in the few- and zero-shot regimes”

Line 160: You say that the model was pre-trained on ImageNet, but the weight file listed in Table 1 references the Cityscapes dataset. Please clarify this. Presumably this means you performed transfer learning, where you used a pretrained model, then finetuned on your labeled dataset?

Line 161: If ten egg images were used for training, how many were used for testing? Any for validation? These are critical details.

Line 171: Rewrite to “In this paper, we used intersection over union (IoU) and pixel accuracy (PA)…” Typically the abbreviation goes in parentheses. Also IoU != “intersection-to-merge”

Integrate Eqs 7 and 8 more naturally into the flow of the writing. They appear after you’ve moved on from the description of IoU and PA.

Line 179: This is not a complete sentence beginning with “where” – could change to “Here, TP…” I would combine these sentences into one, simply listing what each type of error means (TP, FP, etc.)

Results

Line 185: I’m not sure what you mean by “cross-merge ratio”. I would simply say “mean intersection over union” since that is what MIoU means

Line 190: I don’t know what this sentence means: “Therefore, we used the average…”

Table 2: I would say “Reference” instead of “Ref” to be clear

Line 206: I would specify “was consistently less than 1 mm” rather than “is less than 1 mm.” 

Line 208: Please elaborate on this explanation as to why improper segmentation of the reference object may lead to worse measurement of the minor axis specifically. I think I understand your argument, but it would be helpful to spell out your reasoning.

Table 3: Related to my earlier question, are these results for all 30 eggs captured? Doesn’t that mean you evaluated the model on eggs that the segmentation model has already encountered during training? It would be more appropriate to only report results on samples not seen during training in order to properly assess generalization to unseen images and objects. Perhaps only presenting results on the 20 “held-out” eggs would be best.

Discussion

Line 215: “egg s’” should be “eggs’”. Alternatively, since you are always handling a single egg at a time (from what I can tell), you can simply say “egg’s” throughout

Line 221: I don’t quite understand the logical flow of this section. How does this paragraph relate to the previous one or the next? Please make transitions between paragraphs more clear and fully elaborate on each idea. For example, either merge the second paragraph with the third, or elaborate on this idea such that your second paragraph is more than two sentences.

Conclusion

Line 246: “high efficiency of Segformer with zero samples” is a bit confusing since this paper does not adopt zero-shot inference. Are you claiming that Segformer was an appropriate model choice because it is label-efficient (can perform well when only trained on a few labeled examples)? I would make this more clear.

Reviewer 3 Report

First of all, I would like to thank the authors for addressing this issue. The authors propose a specific method for measuring the size of eggs using a single view of a small batch of images. The main goal of this study is to measure the eggs’ major and minor axes based on deep learning and single-view metrology. I have many concerns that I would like the author to explain further. Therefore, major revisions are needed before the paper can be suggested for publication, especially in foods.

Following some suggestions that I hope could be of help:

1.     The author should clearly identify each type of component and add pictures of the actual installation of the system.

2.     The author should state what each figure in Figure 7 is.

3.     Further elaboration on the details of the size, materials, and methods of constructing the egg-carrying components to obtain a single view of the eggs' outline should be provided.

4.     Eggs should be drawn and clearly marked to show how the eggs' major and minor axes are defined.

5.     From Fig. 1, I'm not sure if Fig. 2(c) and (d) are reversed.

6.     I think Figure 9 (b) and (a) are reversed.

7.     Image processing sequence diagrams should be added to this article.

8.     The deep learning process and its details should be explained more clearly.

9.     The authors should clearly state the purpose of development, novel and the strengths of this work.

1.  Please review the similar article and clearly state the innovation of your work in the section of materials and methods and should add a table to compare clearly (accuracy, precision, cost, design innovation, advantages, etc.).

Round 2

Reviewer 1 Report

The quality of the manuscript was not significantly improved.

Reviewer 3 Report

Authors have extensively addressed all the suggestions given during the first round of revision. To my opinion, the paper appears now more complete, the doubts were clarified and thus it is ready to be accepted for publication in Foods.

Author Response

Thank you very much for your help with our manuscript.